# Differences in Fish Abundance in Rivers under the Influence of Open-Pit Gold Mining in the Santiago-Cayapas Watershed, Esmeraldas, Ecuador

Eduardo Rebolledo Monsalve [1], Pedro Jiménez Prado [1], Jon Molinero Ortiz [1] and Theofilos Toulkeridis [2,*]

[1] Escuela de Gestión Ambiental, Pontificia Universidad Católica del Ecuador Sede Esmeraldas, Esmeraldas 080150, Ecuador
[2] Falta de la ESPE, Universidad de las Fuerzas Armadas ESPE, Sangolquí 171103, Ecuador
* Correspondence: ttoulkeridis@espe.edu.ec; Tel.: +59-39-8700-1807

**Abstract:** Illegal gold mining is on the rise in the tropical Andes. The Santiago-Cayapas watershed is located in the north of the Pacific basin of Ecuador, in the Chocó biogeographical region. It is recognized for its high biodiversity, as 62 fish species have been described in the area, and because it contains two of the largest protected areas in the Pacific coast of Ecuador: the mangroves of the Cayapas and Mataje Rivers and the Cotacachi-Cayapas Ecological Reserve. Open-pit gold mining has been described in the area since 2006 and most mining fronts operate illegally and lack any environmental control. Heavy-metal concentrations and fish communities were studied in streams that drain active and abandoned mines, in larger rivers located downstream of the mined areas and in control sites without mining activities. Open-pit mining causes a reduction of dissolved oxygen concentrations and an increase of water temperature, turbidity, and concentrations of Al, Cr, Co, Cu, Fe, Mn, and V. Fish abundance decreased in streams that drain active mines, however, metrics of taxonomic diversity remain unchanged among the study sites. The response of fish communities to open-pit gold mining was complex and driven by the pollution tolerance of each species, the presence of specific adaptions to turbid waters, and changes in the fishing pressure as locals avoid fishing activities in mined areas. Finally, streams that drain abandoned mines showed chemical characteristics, metal concentrations, and fish communities that were similar to control sites, but maintained higher water temperatures than control sites.

**Keywords:** heavy metals; Chocó region; Andean western slopes; illegal gold mining

## 1. Introduction

Mining causes large impacts on the quality of water and sediment in aquatic ecosystems located downstream of mined areas [1–3]. Moreover, mine wastewater discharges have significant effects on aquatic communities, causing structural, functional, and distribution changes in macroinvertebrates, fish, and trophic chains [4–6]. Such impacts may persist decades after the closure of mining facilities [3,7].

Impacts of artisanal gold mining in the tropics are well known and include changes to surface drainage, soil and water contamination with mining wastes, increased water turbidity, reductions in water quality and water for human consumption, losses of natural habitats, biodiversity, and vulnerable species, degradation of fishing grounds, and severe decline of human living conditions and health [5,8–17]. Many studies have also alerted of mercury in water, sediment and fish from water-bodies impacted by auriferous mining in Ecuador [18], Colombia [19–21], Venezuela [22], Peru [23], and Bolivia [24,25].

In the south of Ecuador, small gold mines have existed since the XVth century [26]. In these mining areas, impoverishment of stream benthic communities and high concentrations of toxic metals in water, in sediments, in fish consumed by the local populations, and in the blood and urine of miners and locals with clear negative effects on their moor

skills have been documented [14,27–32]. This knowledge has resulted in proposals to improve mine management and the living conditions of the affected populations [18,26,33]. However, there is less information about the characteristics and the impacts of gold mining in the Santiago-Cayapas watershed and in other areas of Ecuador [34].

The Santiago-Cayapas watershed is located within the Chocó biogeographical region, one of the world biodiversity hotspots that contains an estimate of 6300 species of plants, 830 species of birds, 235 species of mammals, 350 species of amphibians, and 210 species of reptiles [35]. In this watershed, Jiménez-Prado et al. (2015) identified 62 species of freshwater fish, of which 5 are endemic. Gold panning has been a traditional activity of the Afro and Chachi communities in the Santiago-Cayapas watershed, but open-pit gold mining has been documented since 2006 [36]. Most of the mining fronts operate without permits, lack any environmental control, and require the processing of large amounts of sediments for profitability because of the low ore grade. Therefore, open-pit gold mining may lead to large impacts in the rivers and streams of the Santiago-Cayapas watershed, as the amount of waste and its management are determinant of the impacts of mining activities on aquatic ecosystems [37,38]. The aim of this work is to study water quality and fish communities of rivers and streams in the Santiago-Cayapas watershed. Our central hypothesis is that gold mining has triggered changes in fish communities that are related to the presence and proximity of mining activities. It is likely that fish that scavenge on hard substrates or dwell near the streambed, such as Loricaridae, Gobiidae, Eleotridae, Cichlidae, and Heptapteridae, are likely to be more affected by the presence of mining sediments.

## 2. Materials and Methods

### 2.1. Study Area

The Santiago-Cayapas watershed, located in the north of the province of Esmeraldas, has an area of 6321 km$^2$ and discharges into the Pacific Ocean at the southern-most end of the Manglares Cayapas-Mataje Ecological Reserve, the second largest mangrove reserve in Ecuador. The Santiago-Cayapas watershed is the third largest hydrological basin in Ecuador, situated in a seismically active zone [39–43]. The annual rainfall averages 3300 mm, which allows for high water levels throughout the year [44], but discharge is seasonal with higher flows between January and April. The upper part of the watershed belongs to the Cotacachi-Cayapas Ecological Reserve, the largest protected area in Ecuador with 243,638 ha that contains the largest forest patch remaining in the Ecuadorian Chocó (Figure 1).

### 2.2. Mining Activities in the Area

After removing the vegetation and about six to eight meters of the topsoil layer, minerals and metals are dug from square shaped pits of 20–30 m width. The ore, a layer of auriferous grey clay located at six to eight meters depth, is processed in Z-shaped industrial sieves under a high-pressure water flow to wash sediments out and concentrate the gold particles. Because operating gold mines consume about 36 m$^3$ of water per hour, they are always located near rivers, streams, and other water bodies [36]. When the exploitation of a pit finishes, it is abandoned and fills with rain and groundwater and the mining front advances another 20–30 m to start the process over. In 2011, 4889 abandoned ponds were counted in an area of 5709 ha by aerial imagery [36] and that number has likely doubled since then.

### 2.3. Study Sites

Twenty-five sites were selected in the Bogotá, Tululbí, Palabí, Cachaví, Wimbi, Wimbicito, Santiago, and Cayapas rivers and in small tributaries, including the San Antonio, Durango, Comba, Zapallito, María, and Las Antonias streams (Table 1, Figure 1). Sampling sites were grouped into four categories as a function of the presence of mining activities: (1) control sites included streams with no mining activities upstream of the sampling site; (2) active mines included sampling sites located downstream of operating mines; (3) abandoned mines included sites located downstream of mined areas that stopped activity more

than 6 months ago; (4) downstream sites located at least 10 km downstream of any mining activity and receive water from tributaries with and without mining influence.

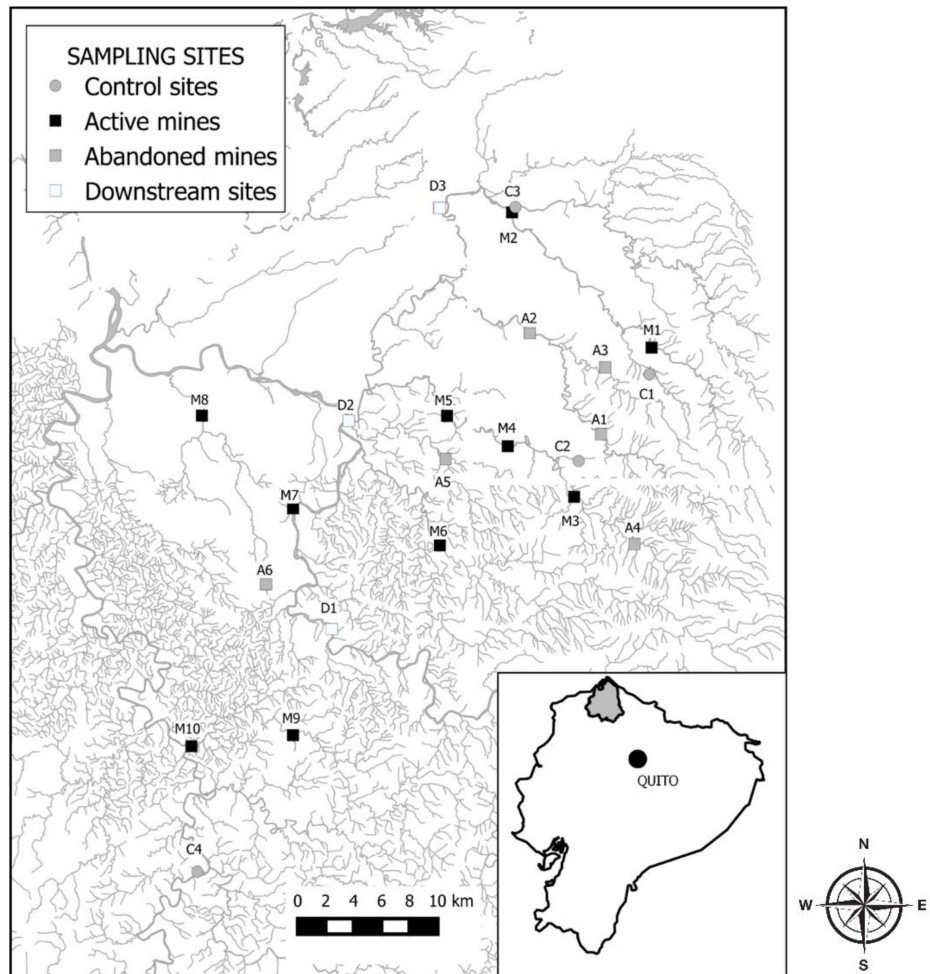

**Figure 1.** Location of the Santiago-Cayapas watershed within Ecuador and study sites (C, control sites; M, active mines; A, abandoned mines; D, downstream sites).

**Table 1.** UTM Coordinates of Location (WGS84, 17N) and characteristics of sampling sites.

| Location | Site | River | UTM X | UTM Y | Order | Elevation (m) | Drainage Area (km²) |
|---|---|---|---|---|---|---|---|
| Control sites | C1 | San José | 762978 | 10118038 | 2 | 155 | 10.3 |
| | C2 | Comba | 757920 | 10111840 | 1 | 120 | 2.5 |
| | C3 | Palabí | 753154 | 10129883 | 4 | 52 | 172.6 |
| | C4 | Cayapas | 730650 | 10082455 | 6 | 45 | 809.3 |
| Active mines | M1 | Tululbí | 763135 | 10119944 | 3 | 108 | 52.4 |
| | M2 | Tululbí | 753154 | 10129883 | 3 | 52 | 131.7 |
| | M3 | Cachabí | 757594 | 10109272 | 4 | 111 | 81.1 |
| | M4 | Cachabí | 752850 | 10112894 | 4 | 72 | 115.6 |
| | M5 | Cachabí | 748501 | 10115067 | 4 | 49 | 131.1 |
| | M6 | Uimbí | 747990 | 10105803 | 5 | 84 | 126.8 |
| | M7 | Las Antonias | 737489 | 10108449 | 1 | 47 | 0.5 |
| | M8 | María | 730966 | 10115077 | 4 | 31 | 70.8 |
| | M9 | Zapallito | 737481 | 10092225 | 3 | 92 | 26.8 |
| | M10 | Zapallito | 730215 | 10091429 | 5 | 40 | 75.4 |

**Table 1.** *Cont.*

| Location | Site | River | UTM X | UTM Y | Order | Elevation (m) | Drainage Area (km$^2$) |
|---|---|---|---|---|---|---|---|
| Abandoned mines | A1 | Bogotá | 759521 | 10113752 | 3 | 114 | 48.9 |
| | A2 | Bogotá | 754426 | 10120966 | 4 | 60 | 109.2 |
| | A3 | Durango | 759827 | 10118532 | 3 | 105 | 21.1 |
| | A4 | San Antonio | 761906 | 10105896 | 2 | 161 | 2.5 |
| | A5 | Uimbicito | 748388 | 10111966 | 3 | 56 | 35.6 |
| | A6 | María | 735549 | 10103020 | 1 | 80 | 2.5 |
| Downstream sites | D1 | Santiago | 740255 | 10099818 | 6 | 76 | 1481.0 |
| | D2 | Santiago | 741474 | 10114714 | 6 | 34 | 1695.4 |
| | D3 | Tululbí | 747964 | 10129930 | 4 | 40 | 428.4 |

*2.4. Field and Laboratory Work*

Study sites were visited 14 times between May 2011 and January 2014 with an irregular schedule to measure water temperature (°C), dissolved oxygen concentration (mg O$_2$L), with a oximeter Mettler Toledo SG-6 Seven Go Pro$^{TM}$ Greifensee, Switzerland; electrical conductivity (μScm), pH, and turbidity (NTU) with the multiparameters Hanna HI9829 Woonsocket, RI, 02895, USA and Wagtech Potalab, Palintest House, Kingsway Team Valley Gateshead Tyne & Wear NE11 ONS United Kindgdom.

In October and December 2013, one-liter water samples were collected, refrigerated in coolers with ice packs, and submitted to an accredited laboratory to measure total concentrations of 20 metals and metalloids by mass spectrometry with inductively coupled plasma according to the EPA 6020A reference method within 48 h of collection. Additionally, fish were sampled by active fishing methods with the help of local fishermen that had the ability to use the fishing gear in a variety of habitats. Two fishermen used 1.25 cm mesh-size cast-nets and two others used a 1 cm mesh-size sweep-net (1.5 × 4 m). At each site, we collected 12 casts with the cast-net and six sweeps with the sweep-net on river turns, banks with submerged vegetation, woody debris accumulations, and shadowed areas. To capture *Chaetostoma marginatum* scold in rocky reaches with fast flowing waters, a cast-net was swept by two fishermen for two minutes along the current.

Captured specimens were refrigerated and identified to species in the laboratory according to Jiménez et al. (2015) [45] and their conservation status was taken from the International Union for Nature Conservation (www.iucnredlist.org (accessed on 10 March 2015)) and Aguirre et al. (2021) [46]. Species richness, absolute and relative species abundances, alpha diversity with the Shannon H index, and the proportion of specimens with deformities were calculated for each sample.

*2.5. Statistical Analyses*

Physical and chemical variables and metal concentrations were fitted to a generalized linear model with the glm function using the location factor (control sites, active mines, abandoned mines, and downstream sites) as the independent variable, the Gaussian error distribution, and the identity link function. One-way ANOVA tables were built with the ANOVA function of the car package [47]. Multiple comparisons by the Tukey test were performed with the emmeans and contrast functions of the emmeans package [48]. To search for patterns among the study sites, a principal component analysis (PCA) was performed with the prcomp function and relevant factors for each axis were selected with the broken-stick method [49]. Prior to these analyses, turbidity and heavy metal concentrations were log-transformed.

Fish, abundances, species richness, and alpha diversity were analyzed with a glm model like the physical and chemical variables. For the fish richness and abundance models, the Poisson error distribution and the identity link function were used. For the alfa diversity

model, the Gaussian error distribution and the identity link function were used. Patterns in the distribution of fish species were analyzed with a non-metric multidimensional analysis (NMDS) with the metaNMDS function of the vegan package [50]. Total fish catch of the two samplings was calculated and converted to relative abundances for the NMDS analysis. Only species that were present at more than two study sites were included in the NMDS analysis. Gradients of species' relative abundances and environmental variables including study site characteristics (Table 1), physicochemical variables, and heavy metal concentrations were added to the NMDS plots with the envfit function. Physicochemical variables and heavy metal concentrations of the two samplings were averaged and log-transformed prior to the NMDS analysis. Watershed area was also log-transformed prior to the NMDS analysis. All statistical analyses were performed in R [51].

## 3. Results

### 3.1. Water Quality

There were large differences in the water physical and chemical characteristics among the study sites (Figure 2, Tables A1 and A2). Water temperature in the streams that drain active and abandoned mines was significantly higher than in the control and downstream sites with an increase of about 1.5 °C ($F_{3,193} = 13.3$, $p < 0.001$). Moreover, streams that drain active mines showed significantly lower oxygen concentrations than the other study sites ($F_{3,172} = 12.6$, $p < 0.001$). Differences in conductivity were only significant between the streams that drain active mines and the control sites ($F_{3,149} = 3.03$, $p < 0.05$). In this case, mean conductivity at the streams that drain active mines almost doubled the value observed at the control sites. There were significant differences in pH among the studied sites ($F_{3,177} = 4.90$, $p < 0.01$), but the differences were negligible and mean pH varied between 6.9 and 7.3 among the sites. Turbidity showed the largest significant differences among the study sites ($F_{3,190} = 4.90$, $p < 0.01$). Mean turbidity in the streams that drain active mines was seven-fold higher than in the control sites and the streams that drain abandoned mines, while it was intermediate in the downstream sites.

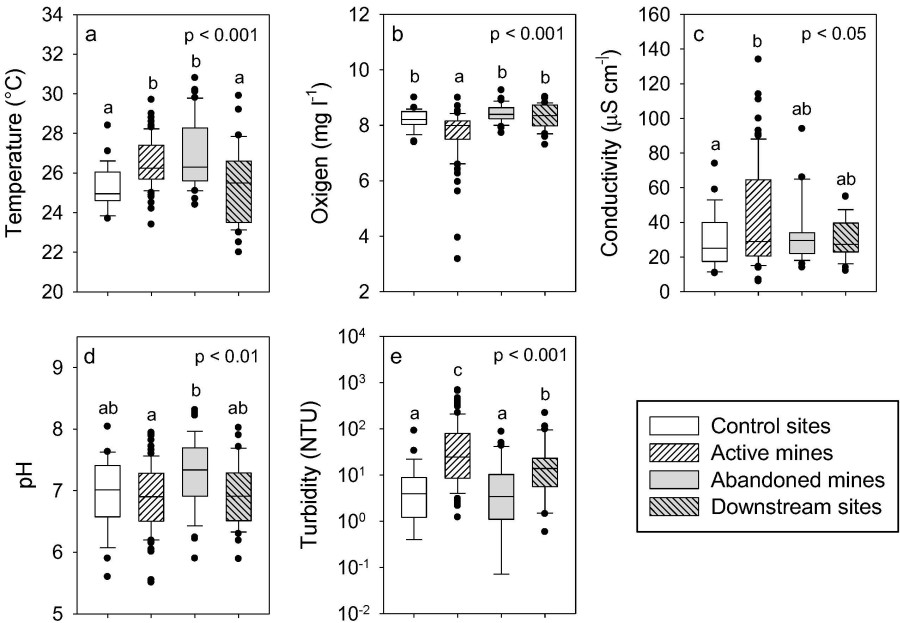

**Figure 2.** Boxplots of physical and chemical variables at the study sites (**a**) = Temperature, (**b**) = Dissolved oxygen, (**c**) = Electrical conductivity, (**d**) = pH and (**e**) = Turbidity. Results of one-factor ANOVAs (location) and multiple comparisons with Tukey test are also shown. There are no significant differences among locations with the same letter. Mean values and statistical analyses are detailed in Tables A1 and A2.

Heavy metal concentrations showed two distinct patterns among the study sites (Figure 3, Tables A1 and A2). The first group showed no significant differences among the study sites and included As, Cd, Se, Sr, and Zn (As: $F_{3,39} = 0.16$, $p > 0.05$; Cd: $F_{3,39} = 1.89$, $p > 0.05$; Mg: $F_{3,39} = 2.70$, $p > 0.05$; Se: $F_{3,39} = 0.25$, $p > 0.05$; Sr: $F_{3,39} = 0.75$, $p > 0.05$; Zn: $F_{3,39} = 0.16$, $p > 0.05$). Additionally, a few metals including Hg, Ag, Tl, and U with concentrations below 0.2 µg L$^{-1}$ in more than 95% of the samples that were not included in Figure 3 also showed the same pattern (Hg: $F_{3,39} = 0.16$, $p > 0.05$; Ag: $F_{3,39} = 0.16$, $p > 0.05$; Tl: $F_{3,39} = 0.16$, $p > 0.05$; U: $F_{3,39} = 0.16$, $p > 0.05$). The second group of metals showed significant differences among the study sites and included Al, Ba, Cr, Co, Cu, Fe, Mn, Ni, Pb, and V (Al: $F_{3,39} = 16.9$, $p < 0.001$; Ba: $F_{3,39} = 7.04$, $p < 0.001$; Cr: $F_{3,39} = 15.1$, $p < 0.001$; Co: $F_{3,39} = 11.9$, $p < 0.001$; Cu: $F_{3,39} = 8.22$, $p < 0.001$; Fe: $F_{3,39} = 12.1$, $p < 0.001$; Mn: $F_{3,39} = 8.03$, $p < 0.001$; Ni: $F_{3,39} = 4.47$, $p < 0.01$; Pb: $F_{3,39} = 3.98$, $p < 0.05$; V: $F_{3,39} = 8.03$, $p < 0.001$). Concentrations of these metals in the streams that drain active mines were higher than in the control sites or in the streams that drain abandoned mines.

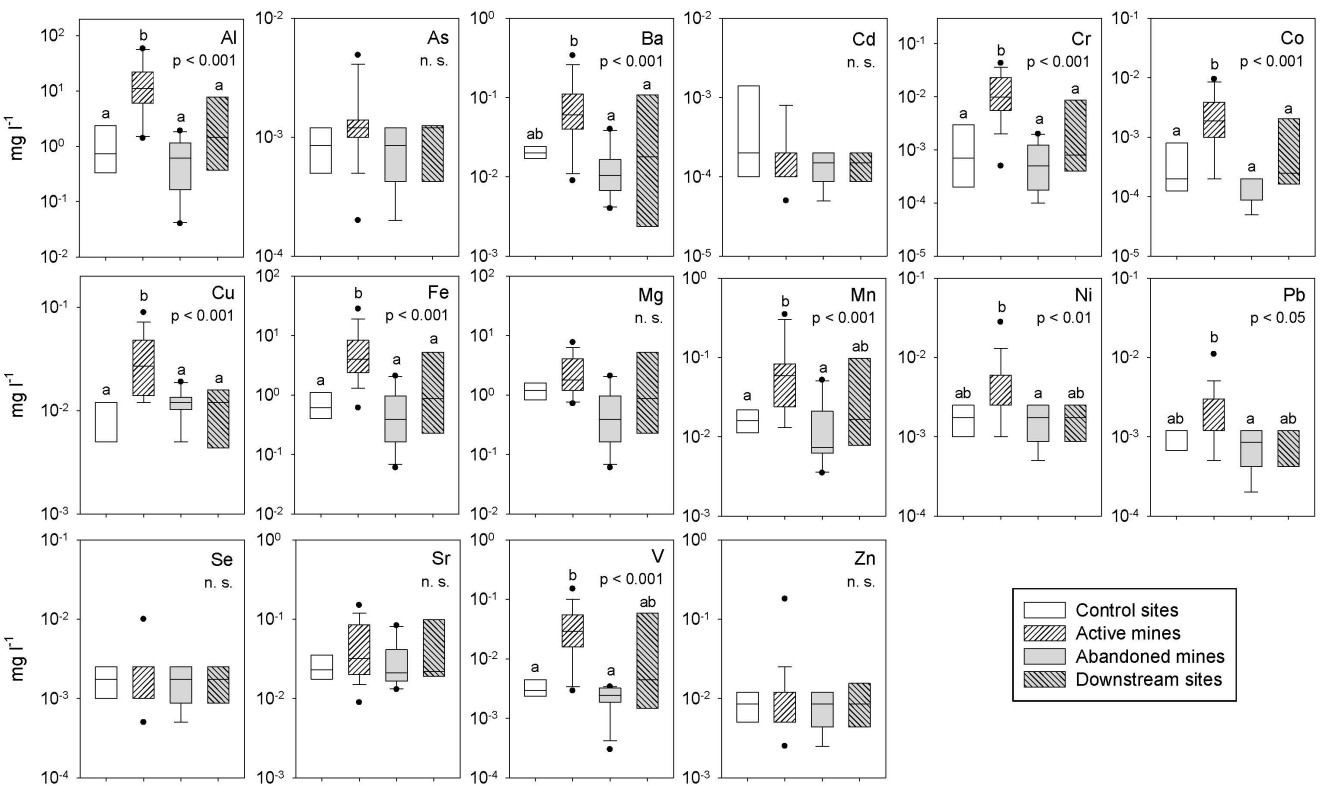

**Figure 3.** Boxplots of heavy metal concentrations at the study sites. Results of one-factor ANOVAs (location) and multiple comparisons with Tukey are also shown. There are no significant differences among locations with the same letter. Mean values and statistical analyses are detailed in Tables A1 and A2.

The two first axes of the PCA analysis explained 60% of the variance in the data (Table 2). The first axis explained 45% of the variance and was positively correlated with turbidity and the concentrations of Al, Co, Cr, St, Fe, Mn, V, and Zn. The second axis explained 15% of the variance and was positively correlated with pH and turbidity and the concentrations of Mg, and negatively correlated with the concentrations of Cd, Se, and Zn. Most samples from streams that drain active mines were located on the right side of the first axis with positive scores, which indicates higher turbidity and concentrations of Al, Co, Cr, St, Fe, Mn, V, and Zn (Figure 4). Samples were segregated depending on the sampling date along the second axis. Most samples collected in December were located on the upper side of the second axis, which indicates higher pH and turbidity and higher

concentrations of Mg, but a dilution of Cd, Se, and Zn concentrations. The first axis of the PCA is showing the impact of mining on water chemistry, while the second axis is showing the effect of rain and higher water discharges at the study sites. Under rainy conditions, the higher dispersion of the samples collected in December along the first axis suggest a stronger impact of active mines on water chemistry.

**Table 2.** Results of the PCA on the physical and chemical variables and heavy metal concentrations. The variance explained by the axes and loadings of variables that were selected by the broken-stick method are listed.

|  | **PC1** | **PC2** |
|---|---|---|
| Variance % | 45% | 15% |
| Cum. Variance % | 45% | 60% |
| Temperature | — | — |
| Oxygen | — | — |
| pH | — | 0.396 |
| Turbidity | 0.226 | 0.209 |
| Conductivity | — | — |
| Al | 0.307 | — |
| As | — | — |
| Ba | — | — |
| Cd | — | −0.316 |
| Co | 0.292 | — |
| Cu | — | — |
| Cr | 0.293 | — |
| St | 0.179 | — |
| Fe | 0.310 | — |
| Mg | — | 0.301 |
| Mn | 0.290 | — |
| Ni | — | — |
| Pb | — | — |
| Se | — | −0.420 |
| V | 0.300 | — |
| Zn | 0.123 | −0.290 |

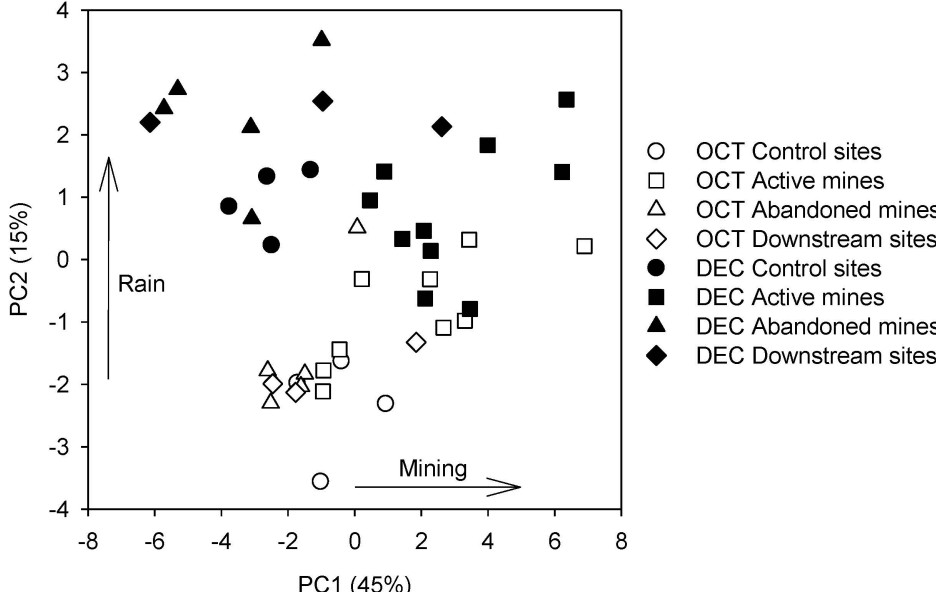

**Figure 4.** Results of the PCA on the physical and chemical variables and heavy metal concentrations at the study sites (OCT, October samples; DEC, December samples, CS, control sites; AcM, active mines; AbM, abandoned mines; DS, downstream sites).

*3.2. Fish Communities*

A total of 33 fish species from 14 families were captured in this study (Table A3). The most abundant families were Characidae, Loricaridae, Cichlidae, and Eliotridae which represented 88% of the 1135 individuals captured. The Characidae predominated in all study sites and represented 57% of the total catch. All species collected were native to Ecuador. Among them, 5 endemic species were captured: *Bryconamericus simus* [52] and *Andinoacara blombergi* [53] that are endemic to the Santiago-Cayapas and Mira rivers; *Sturisoma frenatum* [54] that is endemic to the Santiago-Cayapas, Guayas, and Esmeralda rivers; *Sternopygus arenatus* [55] that is endemic to lowland rivers in the central and northern coast of Ecuador; and *Sycidium rosembergi* [56] that is endemic to the Santiago-Cayapas and Esmeraldas rivers. Of the collected species, 4 appear in the IUNC red list: *Pseudochalceus longianalis* [57] as vulnerable, *S. rosembergi* and *Brycon posadae* [58] as near threatened, and *S. frenatum* as critically endangered, but the remaining 6 species have not been evaluated or lack sufficient data for evaluation.

There were significant differences in fish abundance among the study sites ($F_{3,38} = 3.22$, $p < 0.05$). Fish abundance in streams that drain active mines and in the downstream sites was lower than in the streams that drain abandoned mines and the control sites (Figure 5a). Furthermore, fish abundance in the streams that drain abandoned mines was lower than in the control sites. There were lower abundances of endemic species in streams that drain active mines and in the downstream sites (Figure 5b), but the differences were not significant ($F_{3,38} = 1.61$, $p > 0.05$). The abundance of species cataloged as critically endangered, near threatened, and vulnerable by the IUCN in streams that drain active mines was significantly lower than in control sites and streams that drain abandoned mines ($F_{2,34} = 15.1$, $p < 0.001$). Mean species richness varied between 5 and 6 and mean alpha diversity varied between 1.2 and 1.4 among the studied sites (Figure 5a,c, Tables A4 and A5) and these variables showed no significant differences (richness: $F_{3,38} = 0.43$, $p > 0.05$; alpha diversity: $F_{3,38} = 0.51$, $p > 0.05$).

The most abundant and widely distributed species was *Bryconamericus dahli* [59], which represented almost 40% of the total catch. This species appeared in 21 of the 23 study sites and showed no preferences about the presence or absence of mining activities (Table 3). Seven or 3% of the specimens of *B. dahli* from streams that drain active mines showed large body deformities, but no deformities were observed in the specimens from other study sites. The second most abundant species was *Chaestostoma marginatum* [60], which accounted for 13% of the total catch, but it was absent from the control sites. The last species that was common was *Roeboides occidentalis* [61], which represented 10% of the total catch and showed no preferences for the presence or absence of mining activities. Other species showed relative abundances below 10%. Among them, *A. blombergi* and *P. longianalis* were absent from the streams that drain active mines; *S. arenatus*, *Rineloricaria jubata* [62], and *Astyanax ruberrinus* [63] were absent from the control sites; and *S. frenatum* was absent from streams that drain abandoned mines.

Many species including *P. longianalis*, *S. arenatus*, *Brycon dentex* [64], *S. rosembergi*, *A. banana*, *S. frenatum*, and *Hoplias malabaricus* [65] were absent from the downstream sites. Other species that were present on all the types of study sites were *Hemieleotris latifasciata* [66], *Pimelodella elongata* [64], *Gobiomorus maculatus* [67], and *Pseudocurimata lineopunctata* [68]. Finally, a group of 12 species (*B simus*, *Astyanax festae* [69], *Pseudopoecilia fria* [70], *Eleotris picta* [71], *Pseudophallus starksii* [72], *Lebiasina astrigata* [73], *B. posadae*, *Chaestostoma fischeri* [74], *Hemiancistrus* sp. [75], *Pimelodella* sp. [76], *Strongylura fluviatilis* [73]. and *Andinoacara rivulatus* [64]) that represented less than 5% of the total catch were too scarce to infer any site preference. These species were not listed in Table 3 and were not included in the NMDS analysis.

**Table 3.** Relative abundances of fish species collected at the study sites. Only species that appeared in three or more sites are represented ● > 20%; •, 10–20%; ○, 5–10%; +, < 5%).

| | Control Sites | | | | Abandoned Sites | | | | | | Downstream Sites | | | Mining Sites | | | | | | | | | |
|---|---|---|---|---|---|---|---|---|---|---|---|---|---|---|---|---|---|---|---|---|---|---|---|
| | C1 | C2 | C3 | C4 | A1 | A2 | A3 | A4 | A5 | A6 | D1 | D2 | D3 | M1 | M2 | M3 | M4 | M5 | M6 | M7 | M8 | M9 | M10 |
| *Briconamericus dalhi* | ● | ● | ● | + | ● | ● | ○ | ● | ● | ● | ● | ● | | ● | ● | ● | ● | ● | ● | ● | ● | ● | |
| *Chaestostoma marginatum* | | | | | ● | ● | ● | ● | | | | | | ● | + | ● | ○ | | ● | | | ○ | |
| *Roeboides occidentalis* | | · | | ○ | ● | ○ | | ○ | ● | ● | ○ | | ● | ○ | | ● | | | ● | ● | + | + | |
| *Hemielotris latifasciata* | | | | ● | + | + | ○ | + | | | ● | ● | | ○ | | | + | | + | ○ | ○ | | ● |
| *Pseudochalceus longianalis* | ● | · | | | | | ○ | + | | | | | | | | | | | | | | | |
| *Cichlasoma festae* | + | + | | ○ | ● | ○ | + | + | ● | | | ● | | | | + | | ○ | | | + | | |
| *Pimelodella elongatus* | | | ○ | | | | + | | + | | ○ | | ● | ○ | ● | + | ○ | ● | + | | + | | ● |
| *Gobiomorus maculatus* | | | + | + | + | + | ○ | | | | + | ● | ● | | ● | + | | ● | + | + | + | | |
| *Cichlasoma ornatum* | | | ○ | + | ○ | + | + | + | + | ○ | + | ● | | ○ | | + | + | | | + | + | ● | |
| *Sternopygus arenatus* | | | | | | + | | | | ○ | | | | | + | + | ● | ● | + | | + | ● | ● |
| *Andinocara blombergi* | | + | ● | | + | | + | | | | + | ● | | | | | | | | | | | |
| *Pseudocurimata lineopunctatus* | + | + | | + | | | | + | | | ○ | ● | | | | | | | | | ○ | ○ | |
| *Brycon dentex* | | + | | | + | + | ○ | | | | | | | ○ | | | + | ○ | + | | | + | |
| *Sicydium rosembergi* | | ○ | | | | + | | | | | | | | | + | | | | | | | | |
| *Rinelocaria jubata* | | | | | | | + | + | | | | ○ | | | | + | + | + | | | | + | |
| *Awaous banana* | | | | ○ | + | | | | | | | | | | | | + | | | | | | |
| *Astianax ruberrinus* | | | | | | | ○ | | | | + | | | | | | + | | | ○ | | | |
| *Sturisoma frenatum* | | | | + | | | | | | | | | | | | + | | | | ○ | | ● | |
| *Hoplias malabaricus* | | + | | | | | | | | ○ | | | | | | | | | | | + | | |
| Richness | 5 | 11 | 4 | 9 | 10 | 11 | 12 | 11 | 5 | 6 | 8 | 9 | 5 | 10 | 8 | 10 | 8 | 8 | 10 | 6 | 11 | 8 | 5 |

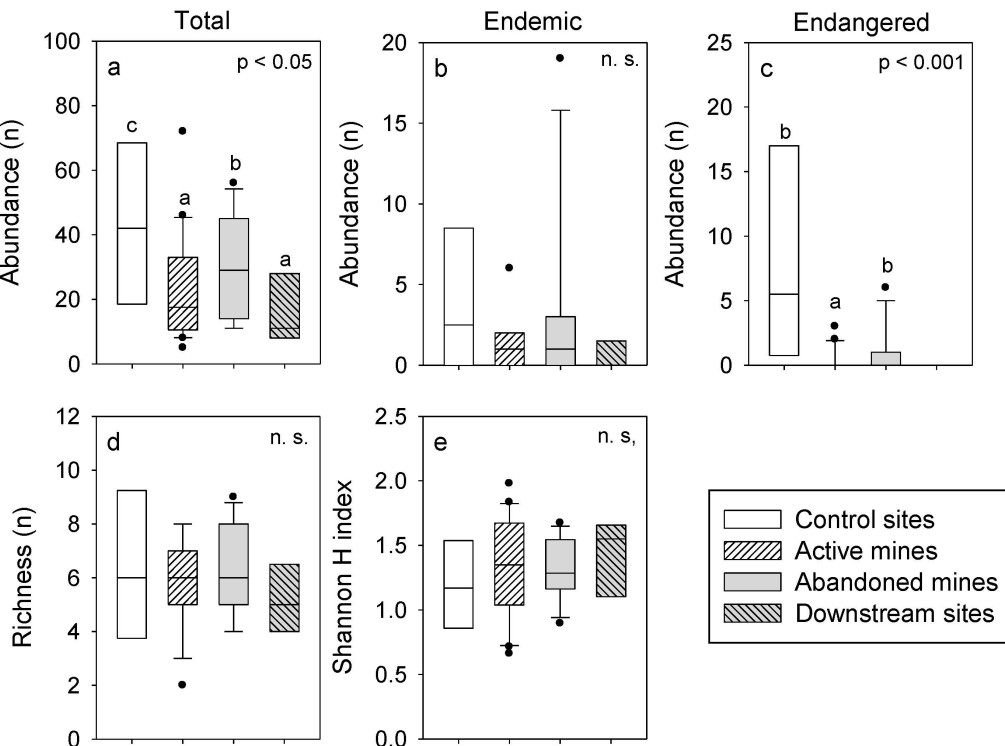

**Figure 5.** Boxplots of fish richness (**d**), abundance (**a**) = total fishes, (**b**) = endemic fishes and (**c**) = endangered fishes, and alpha diversity measured as the Shannon H index (**e**) at the study sites. Results of one-factor ANOVAs (location) and multiple comparisons with Tukey are also shown. There are no significant differences among locations with the same letter. Mean values and statistical analyses are detailed in Tables A4 and A5. ns means not statical differences.

Despite some overlapping, a segregation of the study sites was observed along the first axis of the NMDS analysis (Figure 6a). Control sites were located on the left side of the first NMDS axis, while streams that drain active mines were located on the right side. Also, several antagonistic species were observed along the first NMDS axis. *P. longianalis* (ps1) was associated with control sites, while *S. arenatus* (st1) and *P. elongatus* (pi) were associated with streams that drain active mines. Segregation of study sites along the second NMDS axis was not related to mining activities. The second NMDS axis seems to be related to habitat differences among the study sites as *C. marginatum* (ch) and *B. dentex* (br2), with preference for shallow and fast waters, were associated with positive values along the second NMDS axis, and *P. lineopunctatus* (ps2) and *G. maculatus* (go), with preference for deep and slow waters were associated with negative values. The third axis of the NMDS segregated the control site in the Palabí river from the other study sites (Figure 6c). This site showed a distinct fish community of low richness (4 species) that contained 67% of the captured individuals of *A. blombergi*.

Turbidity and heavy metal concentrations were the main environmental variables related to the gradients of fish abundances along the first NMDS axis (Figure 6b). Further, higher relative abundances of *S. rosembergi* were related to higher elevation of the study sites. There were no environmental variables related to the second NMDS axis, probably because our study contained no metrics related to in-stream habitat. Finally, the control site in the Palabí river is characterized by a higher concentration of Cd than other study sites (Figure 6d).

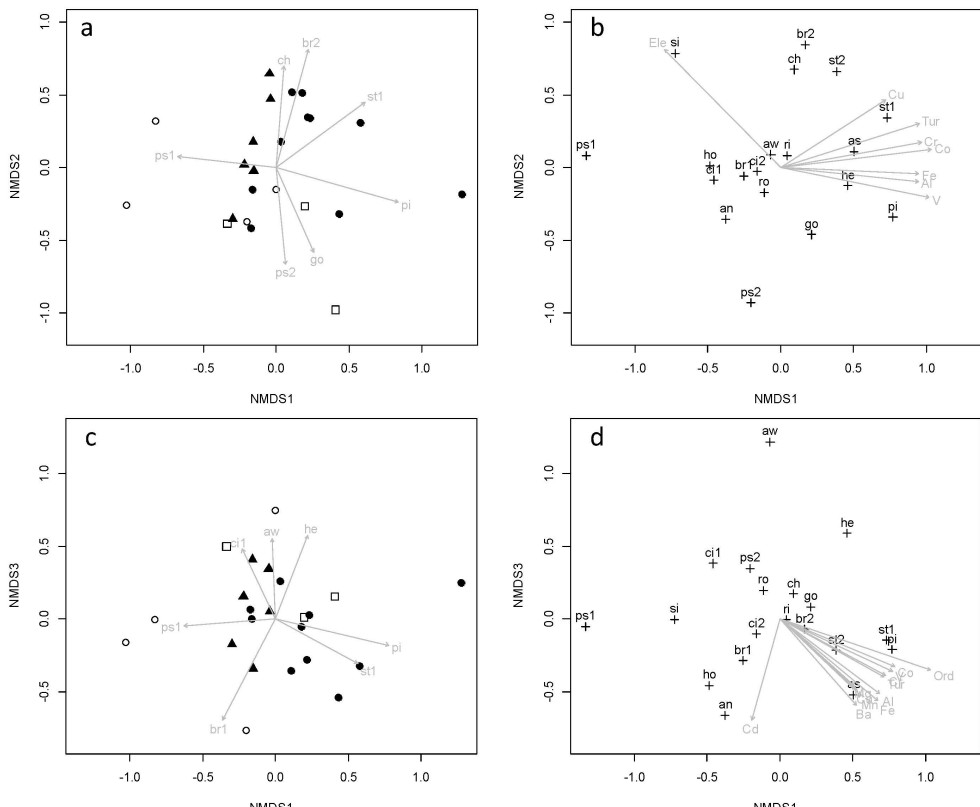

**Figure 6.** Results of NMDS of relative abundances of fish species at the study sites (stress = 0.14; ● control sites; ■, active mines; △, abandoned mines; ◇, downstream sites, empty figures correspond to october and black figures correspond to december). Species included in the analysis were present at more than two study sites: br1, *Briconamericus dalhi*; ch, *Chaestostoma marginatum*; ro, *Roeboides occidentalis*; he, *Hemielotris latifasciata*; ps1, *Pseudochalceus longianalis*; ci1, *Cichlasoma festae*; pi, *Pimelodella elongatus*; go, *Gobiomorus maculatus*; ci2, *Cichlasoma ornatum*; st1, *Sternopygus arenatus*; an, *Andinoacara blombergi*; ps2, *Pseudocurimata lineopunctatus*; br2, *Brycon dentex*; si, *Sicydium rosembergi*; ri, *Rinelocaria jubata*; aw, *Awaous banana*; as, *Astianax ruberrinus*; st2, *Sturisoma frenatum*; ho, *Hoplias malabaricus*. Gradients of environmental variables also shown: Ord, channel order; Ele, elevation; Tur, turbidity.

## 4. Discussion

Although mining activities in northern Esmeraldas are usually referred as artisanal, they cannot be considered artisanal because of the size of the labor force and the volume of processed ore. According to Hammond et al. (2013) [77], they fall between the medium- to large-scale categories and lack appropriate environmental management systems to contain and treat the effluents they produce [44,78]. Bulldozers and retro excavators move large volumes of mine waste and create physical barriers that modify water flow, displacing the natural channel from its original position. Open pits fill with water and miners pump this water or water from nearby rivers for washing the gold ore. Overtopping or collapse of the pit walls and leakages in the pipes that feed the pumps cause the discharge of large volumes of water and fine sediment into rivers and streams.

Most studies about the impacts of gold mining have focused on the detrimental effects of Hg on aquatic organisms, gold miners, and related human populations [16,79], which have led to global regulations on the use of Hg [80]. In the Santiago-Cayapas watershed, gold is found in the form of flakes that are recovered gravimetrically and there is no evidence of Hg or other amalgamating compounds. Nor is there evidence that the removable filters of the Z-shaped sieves are being transported elsewhere for treatment with Hg. As amalgamating compounds are not used in the study area and Hg

concentrations were below the detection level in all the samples, this is an opportunity to observe environmental impacts caused by other changes in water quality, such as the excess of fine sediments and of heavy metals.

Effluents from open-pit gold mining activities in the Santiago-Cayapas watershed have reduced the water quality of the receiving rivers and streams. Streams receiving the effluents of active mines showed significantly higher concentrations of Al, Cr, Co, Cu, Fe, Mn, and V than control sites. A few metals, including Ba, Ni, and Pb, also showed higher concentrations in streams that drain active mines, but the differences with control sites were not significant. Streams that receive the effluents of active mines in the Santiago-Cayapas watershed have lower As and Cu concentrations, similar Cd concentrations, and higher Pb concentrations than in mining areas in the south of Ecuador [28,29,31]. However, concentrations of heavy metals in a mining area of the Amazonas watershed [81] were higher than in impacted streams in the Santiago-Cayapas watershed, except for Al and Fe. Heavy metals released from mining areas pose a potential risk for human populations and the environment [32]. According to the Ecuadorian legislation [82], water from sites that received effluents from active mines surpassed the reference values of Al, Cr, Cu, Fe, Mn, Ni, Pb, and Se for the preservation of aquatic life (Table 4). However, there were also samples from the control sites that surpassed some reference values, so other activities such as agriculture and deforestation are also potential sources of heavy metals in the study area.

The presence of individuals with body deformities is a clear indicator of deleterious effects of mine effluents on aquatic life as there is ample evidence of histopathological damage and deformities in fish exposed to heavy metals [83–87]. However, an evidence approach is required to demonstrate this situation, since there are other pressures in the environment.

**Table 4.** Percentage of samples that comply with reference values for the support of aquatic life according to the Ecuadorian environmental law. Only parameters with established reference values are listed.

|  | Control Sites | Active Mines | Abandoned Mines | Downstream Sites |
|---|---|---|---|---|
| pH | 77 | 79 | 93 | 83 |
| Aluminum (mg $L^{-1}$) | 0 | 0 | 20 | 17 |
| Arsenic (µg $L^{-1}$) | 100 | 100 | 100 | 100 |
| Barium (µg $L^{-1}$) | 100 | 100 | 100 | 100 |
| Cadmium (µg $L^{-1}$) | 75 | 100 | 100 | 100 |
| Chrome (µg $L^{-1}$) | 100 | 84 | 100 | 100 |
| Cobalt (µg $L^{-1}$) | 100 | 100 | 100 | 100 |
| Copper (µg $L^{-1}$) | 38 | 0 | 20 | 33 |
| Iron (mg $1^{-1}$) | 0 | 0 | 40 | 33 |
| Manganese (µg $L^{-1}$) | 100 | 79 | 100 | 83 |
| Mercury (µg $L^{-1}$) | 100 | 100 | 100 | 100 |
| Nickel (µg $L^{-1}$) | 100 | 95 | 100 | 100 |
| Silver (µg $L^{-1}$) | 100 | 100 | 100 | 100 |
| Lead (µg $L^{-1}$) | 25 | 11 | 50 | 33 |
| Selenium (µg $L^{-1}$) | 50 | 47 | 50 | 50 |

Compared to control sites, we also observed higher temperature, conductivity, and turbidity, lower oxygen concentrations, and the formation of thick sediment layers on the stream-bed in streams that drain active mines. Abundance of fish declined in streams that receive effluents from active mines, which indicates that fish avoid the conditions created by mine effluents. Excess of fine sediment is a condition that many fish species avoid seeking refuge in unimpacted reaches, because it reduces food availability and can cause abrasion and clogging of the gill epithelium reducing gas exchange [88–91]. The presence of individuals of endemic and endangered species also decreased in streams that receive effluents from active mines. Therefore, open-pit gold mining is a big threat for endemic

species of reduced distribution already known to be at risk, such as *S. frenatum* (critically endangered) and *S. rosembergi* (near threatened), and for others whose conservation status has not been evaluated, such as *B. simus*. Among the species of ample distribution, it would be advisable to take actions to avoid the local extinction of *P. longianalis* (vulnerable) and *B. posadae* (near threatened) in the Santiago-Cayapas watershed. Mining is considered a major threat to aquatic ecosystems and biodiversity across South America [77,92]. In Ecuador, mining has also been identified as a major threat to freshwater fish in the eastern and western slopes of the Andes, which is exacerbated when mining fronts are illegal and operate without environmental controls [46]. Otherwise, there were no significant differences in species richness or alpha diversity among the study sites. Species richness and other measures of taxonomic diversity have been used to monitor the impacts of mining on fish assemblages with mixed results [5,93–95], but are considered as poor indicators of these impacts [96]. In the Santiago-Cayapas watershed, some species disappear in the streams that receive effluents from active mines, but they are substituted by other species that are pollution tolerant and take advantage of the newly created conditions, so broad metrics of taxonomic composition remain unchanged among the study sites.

Analysis of the impacts of mining on fish communities is based on current knowledge about the distribution and the ecology of freshwater fish in western Ecuador [45]. The most abundant species, *B. dahli* (Characidae), is omnivorous and tolerant to contamination and was present in almost all the study sites. *C. marginatum* (Loricariidae) scavenges on biofilms and was only found in streams that drain active and abandoned mines. This species is attached to rocks and boulders in areas of strong current where sedimentation is limited, so it may tolerate the effluents from active mines. *R. occidentalis* (Characidae) is omnivorous and sometimes found associated to floating garbage and was present in controls and streams that receive effluents from active mines. *P. longianalis* (Characidae) and *A. blombergi* (Cichlidae) were absent from streams that receive effluents from active mines. *P. longianalis* is found in coastal rivers of Ecuador and Colombia associated to gravel and bedrock bottoms in small- to medium-sized channels with clear water. *A. blombergi* is endemic to the Santiago-Cayapas and Esmeraldas watersheds, but there is not much information about its biology and ecology. Our study suggests that both species can be considered as indicators of good water quality in the study area. On the contrary, *S. arenatus* (Sternopygidae) and *P. elongatus* (Heptapteriadae) were mainly found in streams that receive the effluents of active mines. Both species are predators that are active at night and use electrical sensors (*S. arenatus*) and antennae (*P. elongatus*) to detect their prey. A few other species showed no preference for control or impacted streams, but responded to some other environmental variables. *S. rosembergii* appeared mainly in streams located at an elevation of 100 m or higher, so this threatened species may have found refuge on higher grounds of the Santiago-Cayapas watershed outside the mined areas. A group of species including *C. festae*, *G. maculatus*, and *P. lineapunctatus* were more abundant in downstream sites, which indicates a preference for larger streams.

In opposition to our initial hypothesis that prioritized habitat preferences to explain the impact of mining on fish, the response of fish communities to environmental changes caused by open-pit gold mining was complex and driven by the pollution tolerance of each species, the presence of specific adaptions to turbid waters, and changes in the fishing pressure as locals avoid or reduce fishing activities in mined areas. It has been observed that functional composition of fish assemblages is more stable than taxonomic composition in streams impacted by mines [96,97]. On the contrary, functional adaptations play a major role to explain the impact of mining on the distribution of fish in the Santiago-Cayapas watershed. Night predators that forage using tactile and olfactory senses are favored in the turbid waters of streams that drain active mines [88]. Prevalence of night predators has also been observed in streams impacted by gold mining in Suriname [13], so it could be a typical response of fish assemblages to increasing turbidity of the Chocó and other regions. Furthermore, fish species of ample distribution in the watershed and known to be pollution tolerant are found in streams that drain active mines. The distribution of the population

near rivers in the Cayapas-Santiago watershed and their dependence on fishing for survival also influences the response of fish communities to mining. Population is relatively high in the study area and miners operate near population centers, sometimes surrounding them and even isolating some houses from others. Locals avoid fishing in streams that drain active mines, either because they are suspicious about the pollution caused by mine effluents or because they recognize the lower fish abundances. Therefore, streams that drain active and abandoned mines become a refuge for *C. marginatum*, a species that is highly appreciated and actively fished elsewhere.

In contrast with some studies that found impacts that persist for decades after the termination of mining activities [3,7], we observed that streams that drain abandoned mines have chemical characteristics, metal concentrations, and fish communities that are similar to control sites but maintained higher water temperatures than control sites. Higher temperatures in streams of mined areas are caused by the removal of the riparian vegetation on the banks during the excavation of the pits, as riparian vegetation regulates stream temperature through shading [98,99]. Legacy impacts from mining activities in the Cayapas-Santiago watershed were non-existent or were difficult to detect within the scope and time limitations of this study.

The use of active fishing methods may lead to some uncertainty about the observed differences in fish communities among the study sites, because fishing gear is more likely to be undetected by fish in turbid waters. However, the lower fish catches obtained in streams that receive mine effluents suggest that the collaboration of expert local fishermen in the study may have overcome any potential bias caused by the methodology.

Finally, some limitations of our study should be highlighted. First, some study sites could not be visited continuously because of unfordable rivers or to ensure the safety of the monitoring team. The lack of easy access to the study sites, which is further complicated during rainy periods, and the violence in the study area limit the opportunities to establish a monitoring program and some impacts may have been undetected. Despite this scenario it is necessary to expand studies towards other effects associated with water quality, such as microbiological analysis and the bioaccumulation of metal both in fish and in inhabitants who consume water from the rivers north of Esmeraldas; however, the analysis capacity in Ecuador to these purposes is limited.

## 5. Conclusions

Illegal open-pit gold mining in the north of Esmeraldas reduces water quality, restricts fish habitats, and modifies the fish assemblages in rivers and streams of the Santiago-Cayapas watershed. Mining causes a reduction of dissolved oxygen concentrations and an increase of water temperature, turbidity, and concentrations of some heavy metals. These changes are also observed in larger rivers located downstream of the mining areas.

The presence of body deformities in fish is probably related to exposure to heavy metals from mine effluents, requiring an evidence approach with specific studies to confirm this situation as there are other sources of pollution in the area such as the abuse of agrochemicals. Two species, *P. longianalis* (Characidae) and *A. blombergi* (Cichlidae), can be considered as bioindicators of good water quality in the study area. Nocturnal predator species that do not rely on visual prey detection increase their presence and activity in mined areas.

In general, most fish species avoid the conditions caused by mine effluents, so open-pit gold mining reduces habitat availability and poses a risk to those species already threatened or endemic to the Santiago-Cayapas watershed. Otherwise, most rivers and streams seem to recover after mining activities cease, although the difficulty of establishing a monitoring program in the study area may mean that some effects caused by open-pit gold mining have been undetected in this study.

**Author Contributions:** Conceptualization, E.R.M. and P.J.P.; methodology, E.R.M. and J.M.O.; software, P.J.P.; validation, E.R.M., J.M.O. and P.J.P.; formal analysis, E.R.M. and J.M.O.; investigation, P.J.P. and J.M.O.; resources, E.R.M. and J.M.O.; data curation, E.R.M.; writing—original draft preparation, T.T. and E.R.M.; writing—review and editing, T.T.; visualization, T.T.; supervision, E.R.M.; project administration, J.M.O.; funding acquisition, E.R.M. and P.J.P. All authors have read and agreed to the published version of the manuscript.

**Funding:** This work has been financed by the Programa de Reparación Ambiental y Social (PRAS) del Ministerio del Ambiente Agua y Transicion Ecologica del Ecuador (MAATE) contracts 064 (2011) and 048 (2013), the Apostolic Vicariate of Esmeraldas donations, and the Pontificia Universidad Católica del Ecuador Sede Esmeraldas internal funds (PUCESE).

**Data Availability Statement:** Not applicable.

**Acknowledgments:** We thank the Pontificia Universidad Católica del Ecuador Sede Esmeraldas (PUCESE) and the Universidad de las Fuerzas Armadas ESPE for their technical support.

**Conflicts of Interest:** The authors declare no conflict of interest.

## Appendix A

**Table A1.** Mean, standard deviation and range of variation of physical and chemical variables and heavy metal concentrations at the study sites.

| | Control Sites | | Active Mines | | Abandoned Mines | | Downstream Sites | |
|---|---|---|---|---|---|---|---|---|
| | Mean ± SD | Min–Max | Mean ± SD | Min–Max | Mean ± SD | Min–Max | Mean ± SD | Min–Max |
| Temperature (°C) | 25.2 ± 1.1 | 23.7–28.4 | 26.5 ± 1.2 | 23.4–29.7 | 26.9 ± 1.7 | 24.4–30.8 | 25.3 ± 1.9 | 22–0–29.9 |
| Oxygen (mg $L^{-1}$) | 8.2 ± 0.4 | 7.4–9.0 | 7.7 ± 0.9 | 3.2–9.0 | 8.4 ± 0.3 | 7.7–9.3 | 8.3 ± 0.4 | 7.3–9.0 |
| Conductivity ($\mu S\,cm^{-1}$) | 29.5 ± 15.7 | 10.7–74.0 | 42.5 ± 29.5 | 6.0–134.0 | 33.9 ± 19.7 | 14.0–94.0 | 30.1 ± 11.5 | 12.0–55.0 |
| pH | 7.0 ± 0.6 | 5.6–8.0 | 6.9 ± 0.5 | 5.5–7.9 | 7.3 ± 0.6 | 5.9–8.3 | 7.0 ± 0.5 | 5.9–8.0 |
| Turbidity (NTU) | 9 ± 18 | 0–92 | 75 ± 130 | 0–690 | 11 ± 18 | 0–87 | 30 ± 45 | 0–222 |
| Aluminum (mg $L^{-1}$) | 1.2 ± 1.2 | 0.2–3.3 | 15.8 ± 16.6 | 1.4–58.0 | 0.7 ± 0.6 | 0.0–1.9 | 3.4 ± 4.1 | 0.1–10.0 |
| Arsenic ($\mu g\,L^{-1}$) | 0.9 ± 0.4 | 0.5–1.2 | 1.5 ± 1.2 | 0.2–4.9 | 0.8 ± 0.4 | 0.2–1.2 | 1.0 ± 0.5 | 0.2–1.4 |
| Barium ($\mu g\,L^{-1}$) | 20 ± 7 | 8–34 | 90 ± 85 | 9–340 | 14 ± 11 | 4–40 | 63 ± 113 | 1–290 |
| Cadmium ($\mu g\,L^{-1}$) | 0.9 ± 1.6 | 0.1–4.6 | 0.3 ± 0.2 | 0.1–0.8 | 0.1 ± 0.1 | 0.1–0.2 | 0.1 ± 0.1 | 0.1–0.2 |
| Chrome ($\mu g\,L^{-1}$) | 2.0 ± 2.7 | 0.2–8.0 | 32.2 ± 21.9 | 12.0–89.0 | 0.8 ± 0.7 | 0.1–2.0 | 3.4 ± 4.5 | 0.1–10.0 |
| Cobalt ($\mu g\,L^{-1}$) | 0.4 ± 0.5 | 0.1–1.4 | 2.7 ± 2.6 | 0.2–9.5 | 0.2 ± 0.1 | 0.1–0.2 | 1.0 ± 1.5 | 0.1–4.0 |
| Copper ($\mu g\,L^{-1}$) | 9.1 ± 4.9 | 0.5–14.0 | 32.2 ± 21.9 | 12.0–89.0 | 11.7 ± 4.2 | 5.0–19.0 | 11.8 ± 8.5 | 2.5–27.0 |
| Iron (mg $L^{-1}$) | 0.9 ± 0.7 | 0.3–2.5 | 6.7 ± 6.8 | 0.6–28.0 | 0.6 ± 0.7 | 0.1–2.1 | 2.4 ± 3.3 | 0.0–8.4 |
| Magnesium (mg $L^{-1}$) | 1.2 ± 0.5 | 0.7–2.1 | 2.6 ± 2.0 | 0.7–7.7 | 1.4 ± 0.9 | 0.6–3.4 | 1.8 ± 1.0 | 1.1–3.8 |
| Manganese ($\mu g\,L^{-1}$) | 20 ± 14 | 8–52 | 88 ± 97 | 13–350 | 15 ± 17 | 4–52 | 51 ± 72 | 6–190 |
| Mercury ($\mu g\,L^{-1}$) | 0.14 ± 0.05 | 0.10–0.20 | 0.14 ± 0.06 | 0.05–0.20 | 0.14 ± 0.07 | 0.05–0.20 | 0.14 ± 0.07 | 0.05–0.20 |
| Nickel ($\mu g\,L^{-1}$) | 1.8 ± 0.8 | 1.0–2.5 | 5.3 ± 6.4 | 1.0–28.0 | 1.7 ± 0.9 | 0.5–2.5 | 1.7 ± 0.9 | 0.5–2.0 |
| Silver ($\mu g\,L^{-1}$) | 0.55 ± 0.84 | 0.10–2.50 | 0.14 ± 0.06 | 0.05–0.20 | 0.14 ± 0.07 | 0.05–0.20 | 0.14 ± 0.07 | 0.05–0.20 |
| Lead ($\mu g\,L^{-1}$) | 1.4 ± 1.1 | 0.5–4.0 | 2.3 ± 2.5 | 0.5–11.0 | 0.8 ± 0.4 | 0.2–1.2 | 0.9 ± 0.4 | 0.2–1.2 |
| Selenium ($\mu g\,L^{-1}$) | 1.8 ± 0.8 | 1.0–2.5 | 2.2 ± 2.1 | 0.5–10.0 | 1.7 ± 0.9 | 0.5–2.5 | 1.7 ± 0.9 | 0.5–2.5 |
| Strontium ($\mu g\,L^{-1}$) | 28 ± 13 | 16–55 | 49 ± 42 | 9–150 | 31 ± 23 | 13–83 | 67 ± 105 | 19–280 |
| Thallium ($\mu g\,L^{-1}$) | 0.15 ± 0.05 | 0.10–0.20 | 0.14 ± 0.06 | 0.05–0.20 | 0.14 ± 0.07 | 0.05–0.20 | 0.14 ± 0.07 | 0.05–0.20 |
| Uranium ($\mu g\,L^{-1}$) | 0.15 ± 0.05 | 0.10–0.20 | 0.22 ± 0.17 | 0.05–0.70 | 0.14 ± 0.07 | 0.05–0.20 | 0.14 ± 0.07 | 0.05–0.20 |
| Vanadium ($\mu g\,L^{-1}$) | 3.8 ± 2.4 | 2.1–9.3 | 40.5 ± 38.8 | 2.9–150.0 | 2.4 ± 1.0 | 0.3–3.4 | 30.4 ± 54.9 | 1.1–140.0 |
| Zinc ($\mu g\,L^{-1}$) | 20 ± 35 | 5–106 | 19 ± 39 | 3–180 | 8 ± 4 | 3–12 | 10 ± 9 | 3–26 |

**Table A2.** Results of one-way ANOVA (location) and multiple comparisons with Tukey's test of physical and chemical variables and heavy metal concentrations at the study sites (CS, control sites; AcM, active mines; AbM, abandoned mines; DS, downstream sites; n. s., not significant). There are no significant differences between locations with the same superscript.

| Metal | F Value | *p* | Multiple Comparisons |
|---|---|---|---|
| Temperature | $F_{3,193}$ = 13.3 | $p < 0.001$ | CS [a] DS [a] AcM [b] AbM [b] |
| Oxygen | $F_{3,172}$ = 12.6 | $p < 0.001$ | AcM [a] CS [b] DS [b] AbM [b] |
| Conductivity | $F_{3,149}$ = 3.03 | $p < 0.05$ | CS [a] DS [ab] AbM [ab] AcM [b] |
| pH | $F_{3,177}$ = 4.90 | $p < 0.01$ | AcM [a] DS [ab] CS [ab] AbM [b] |
| Turbidity | $F_{3,190}$ = 19.9 | $p < 0.001$ | CS [a] AbM [a] DS [b] AcM [c] |
| Aluminum | $F_{3,39}$ = 16.9 | $p < 0.001$ | AbM [a] CS [a] DS [a] AcM [b] |
| Arsenic | $F_{3,39}$ = 0.16 | n. s. | — |

**Table A2.** *Cont.*

| Metal | F Value | p | Multiple Comparisons |
|---|---|---|---|
| Barium | $F_{3,39} = 7.04$ | $p < 0.001$ | AbM [a] DS [a] CS [ab] AcM [b] |
| Cadmium | $F_{3,39} = 1.89$ | n. s. | — |
| Chrome | $F_{3,39} = 15.1$ | $p < 0.001$ | AbM [a] CS [a] DS [a] AcM [b] |
| Cobalt | $F_{3,39} = 11.9$ | $p < 0.001$ | AbM [a] CS [a] DS [a] AcM [b] |
| Copper | $F_{3,39} = 8.22$ | $p < 0.001$ | CS [a] DS [a] AbM [a] AcM [b] |
| Iron | $F_{3,39} = 12.1$ | $p < 0.001$ | AbM [a] CS [a] DS [a] AcM [b] |
| Magnesium | $F_{3,39} = 2.70$ | n. s. | — |
| Manganese | $F_{3,39} = 8.03$ | $p < 0.001$ | AbM [a] CS [a] DS [ab] AcM [b] |
| Mercury | $F_{3,39} = 0.08$ | n. s. | — |
| Nickel | $F_{3,39} = 4.47$ | $p < 0.01$ | AbM [a] CS [ab] DS [ab] AcM [b] |
| Silver | $F_{3,39} = 2.35$ | n. s. | — |
| Lead | $F_{3,39} = 3.98$ | $p < 0.05$ | AbM [a] CS [ab] DS [ab] AcM [b] |
| Selenium | $F_{3,39} = 0.25$ | n. s. | — |
| Strontium | $F_{3,39} = 0.75$ | n. s. | — |
| Thallium | $F_{3,39} = 0.15$ | n. s. | — |
| Uranium | $F_{3,39} = 1.31$ | n. s. | — |
| Vanadium | $F_{3,39} = 8.03$ | $p < 0.001$ | AbS [a] CS [a] DS [ab] AcM [b] |
| Zinc | $F_{3,39} = 0.50$ | n. s. | — |

**Table A3.** Taxonomical classification of the collected fish specimens (Det, detritivore; Pre, predator; Ins, insectivorous; Omn, omnivorous; Herb, herbivorous; LC, least concern; NE, not evaluated; DD, data deficient; VU, vulnerable; NT, near threatened; CR, critically endangered).

| Family | Species | Type | Feeding | IUNC Status |
|---|---|---|---|---|
| Curimatidae | *Pseudocurimata lineopunctata* [68] | Native | Det | LC |
| Erythrinidae | *Hoplias malabaricus* [65] | Native | Pre | LC |
| Lebiasinidae | *Lebiasina astrigata* [73] | Native | Ins | LC |
| Characidae | *Astyanax festae* [69] | Native | Omn | NE |
| | *Astyanax ruberrinus* [63] | Native | Omn | NE |
| | *Bryconamericus dahli* [59] | Native | Omn | LC |
| | *Bryconamericus simus* [69] | Endemic | — | DD |
| | *Pseudochalceus longianalis* [57] | Native | — | VU |
| | *Roeboides occidentalis* [61] | Native | Omn | LC |
| Bryconidae | *Brycon dentex* [64] | Native | Omn | LC |
| | *Brycon posadae* [58] | Native | — | NT |
| Heptapteridae | *Pimelodella longate* [64] | Native | Omn | LC |
| | *Pimelodella* sp [76] | — | — | — |
| Loricariidae | *Chaestostoma fischeri* [74] | Native | Herb | LC |
| | *Chaestostoma marginatum* [60] | Native | Herb | LC |
| | *Hemiancistrus* sp [75] | — | — | — |
| | *Rineloricaria jubata* [54] | Native | Herb | LC |
| | *Sturisoma frenatum* [54] | Endemic | Herb | CR |
| Sternopygidae | *Sternopygus arenatus* [55] | Endemic | Pre | NE |
| Poeciliidae | *Pseudopoecilia fria* [70] | Native | Ins | LC |
| Belonidae | *Strongylura fluviatilis* [73] | Native | Pre | NE |
| Syngnathidae | *Pseudophallus starksii* [72] | Native | — | LC |
| Cichlidae | *Andinoacara blombergi* [53] | Endemic | Pre | LC |
| | *Andinoacara rivulatus* [64] | Native | Omn | NE |
| | *Cichlasoma ornatus* [100] | Native | Pre | LC |
| | *Cichlasoma festae* [56] | Native | Omn | NE |
| | *Andinoacara rivulatus* [64] | Native | Ins | NE |
| Eleotridae | *Gobiomorus maculatus* [67] | Native | Pre | LC |
| | *Eleotris picta* [71] | Native | Pre | LC |
| | *Hemieleotris latifasciata* [66] | Native | Ins | LC |
| Gobiidae | *Awaous banana* [101] | Native | Herb | LC |
| | *Sycidium rosembergi* [56] | Endemic | — | NT |

**Table A4.** Mean, standard deviation, and range of variation of fish richness, abundance, and alpha diversity at the study sites.

| | Control Sites | | Active Mines | | Abandoned Mines | | Downstream Sites | |
|---|---|---|---|---|---|---|---|---|
| | Mean ± SD | Min–Max | Mean ± SD | Min–Max | Mean ± SD | Min–Max | Mean ± SD | Min–Max |
| Abundance (count) | 45 ± 28 | 17–88 | 22 ± 16 | 5–72 | 31 ± 16 | 11–56 | 17 ± 11 | 7–31 |
| Endemic (count) | 4 ± 4 | 0–10 | 1 ± 2 | 0–6 | 3 ± 6 | 0–19 | 1 ± 1 | 0–2 |
| Endangered (count) | 8 ± 9 | 0–23 | 0 ± 1 | 0–3 | 1 ± 2 | 0–6 | — | — |
| Richness (count) | 6 ± 3 | 3–10 | 6 ± 2 | 2–8 | 6 ± 2 | 4–9 | 5 ± 1 | 4–7 |
| Alpha diversity | 1.2 ± 0.4 | 0.4–1.7 | 1.3 ± 0.4 | 0.7–2.0 | 1.3 ± 0.2 | 0.9–1.7 | 1.4 ± 0.3 | 0.9–1.7 |

**Table A5.** Results of one-way ANOVA (location) and multiple comparisons with Tukey's test of fish richness, abundance, and alpha diversity (CS, control sites; AcM, active mines; AbM, abandoned mines; DS, downstream sites; n. s., not significant). There are no significant differences between locations with the same superscript.

| Variable | F Value | $p$ | Multiple Comparisons |
|---|---|---|---|
| Abundance | $F_{3,38} = 3.22$ | $p < 0.05$ | DS [a] AcM [a] AbM [b] CS [c] |
| Endemic | $F_{3,38} = 1.61$ | n. s. | — |
| Endangered | $F_{2,34} = 15.1$ | $p < 0.001$ | AcM [a] AbM [b] CS [b] |
| Richness | $F_{3,38} = 0.43$ | n. s. | — |
| Alpha diversity | $F_{3,38} = 0.51$ | n. s. | — |

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
