# Peer review of "Differences in Fish Abundance in Rivers under the Influence of Open-Pit Gold Mining in the Santiago-Cayapas Watershed, Esmeraldas, Ecuador"

_water, doi:10.3390/w14192992_

Round 1

Reviewer 1 Report

The present manuscript addresses the impact of gold mining on water quality and fish communities in the Santiago-Cayapas watershed.  The paper is a competent study raising some interesting knowledge and can be accepted after minor changes given below

1. The title seems good, and the abstract also seems to be good. Please add one more introductory line of your objective at beginning of the abstract.

2.  Please correct Keywords (check punctuations)

3. Research gap should be delivered in a clearer way with the directed necessity for future research work.

4. Please provide space between numbers and units. 

5. Result and discussion seem ok. No comment

6. Conclusion looks like a summary. Make it more comprehensive and add future perspectives and mitigatory measures to cope with the situation.

Author Response

Response to reviewer #1

Dear expert reviewer,

As authors of the manuscript now entitled “Impacts of open-pit gold mining on water quality and fish assemblages in the Santiago-Cayapas watershed, Esmeraldas, Ecuador”, we appreciated a lot your description and comments on the document, as we are certain and convinced, that they have been useful to eliminate obvious errors and to enrich the fluency and clarity of the entire article. Hereunder, we will reply and accomplish to change each mentioned point based on your input.

  1. The title seems good, and the abstract also seems to be good. Please add one more introductory line of your objective at beginning of the abstract.

A very nice observation, which has let us to change towards a new title and description:  Impacts of open-pit gold mining on water quality and fish assemblages in the Santiago-Cayapas watershed, Esmeraldas, Ecuador.  The assemblages imply distribution, abundance and richness, the communities involve much more analysis.

We have added an introductory line at the abstract

  1. Please correct Keywords (check punctuations)

Two terms being Pacific basin and Ecuador have been removed and the punctuation has been revised 

  1. Research gap should be delivered in a clearer way with the directed necessity for future research work.

Research is restricted in marginalized territories, therefore it remains difficult to access any funds while the country´s installed capacity for chemical analysis is weak. Illegal mining is a topic that seeks to make itself invisible. As a team, it was decided to avoid these comments that can be understood as subjective, but we added a somewhat homologous idea at the end of the discussion

  1. Please provide space between numbers and units.

The factors of the anovas mentioned in the text were written as subscripts to improve their visualization

  1. Result and discussion seem ok. No comment

Thank you

  1. Conclusion looks like a summary. Make it more comprehensive and add future perspectives and mitigatory measures to cope with the situation.

As you can see, we improved somehow this part

Once again and with all due respect, we are very thankful for your comments and corrections, which helped to see a few unclear parts and or even faults of our side within our manuscript. With your comments we were able to smooth the text, clarify missing parts or wrong spellings, which resulted to a much better than the initial version of this current study.

Thanks a lot on behalf of all authors

Reviewer 2 Report

in the tile open-pit gold mining could be revised for gold placer mining.

Author Response

Response to reviewer #2

Dear expert reviewer,

As authors of the manuscript now entitled “Impacts of open-pit gold mining on water quality and fish assemblages in the Santiago-Cayapas watershed, Esmeraldas, Ecuador”, we appreciated a lot your description and comments on the document, as we are certain and convinced, that they have been useful to eliminate obvious errors and to enrich the fluency and clarity of the entire article. Hereunder, we will reply and accomplish to change each mentioned point based on your input.

in the tile open-pit gold mining could be revised for gold placer mining.

Gold placers refer more to the availability of quantity of gold that would exist in a sector with or without mining activity it´s a kind of gold reserve, our publication refers to the impacts of the activity already declared. Nonetheless, for an other reason we changed improving slightly the previous title of our manuscript.

Once again and with all due respect, we are very thankful for your comments and corrections, which helped to see a few unclear parts and or even faults of our side within our manuscript. With your comments we were able to smooth the text, clarify missing parts or wrong spellings, which resulted to a much better than the initial version of this current study.

Thanks a lot on behalf of all authors

Reviewer 3 Report

The authors have made an effort to evaluate in great detail the physico-chemical properties of water in mine-affected streams. Congratulations, well done.

However, this part of the title "Impact of open-pit gold mining on fish communities'', that part of the title is not confirmed by the text of the study.

The title about the impact of open-pit mining on fish is very brave, but based on this data it is not possible to say with certainty. This opens the door to speculation, which cannot be the basis for scientific arguments.

Has a microbiological analysis of the water samples been performed?

Considering the fact that this is an area exposed to rainwater, bacteria and the pathogens that cause fish diseases can easily be washed away.

Has the health status of the fish been examined?

Has the autopsy of the fish been done?

Do the authors have data on the concentrations of metals and metallothioneins in fish?

There is a very good statement on page 9: "There were significant differences in fish abundance among study sites."

Accordingly, the correct title would be:

Differences in fish abundance in rivers under the influence of open-pit gold mining in the Santiago-Cayapas basin, Esmeraldas, Ecuador.

Special remarks.

Authors must cite literature according to the instructions for authors, consistently throughout the text. So in the case on page 2, "Jimenez-Prado"...or page 13, "Hammond et al. (2013)''

Use of punctuation, such as a period after 2006 (36) on page 2.

Spacing between words, such as on the page 2 ''(37, 38). The goal''..., or page 13 ''(57,44)''

Writing international units of measure m3 and not m3, mg O2L-1 and not mg O2l-1; µScm-1, not µScm-1, etc.

Page 9, ‘’Under then’’ in ‘’Under them’’.

The conclusions must be written in accordance with the facts from the paper. We cannot conclude that mining has reduced habitats, but that it has reduced the quality of those habitats. Based on the above, we cannot exclusively say that mines have changed fish communities? Because we have no data on the health status of the fish.

Are body deformities in fish the only result of the effect of heavy metals on the fish organism?

How can we tell if there are no data on the concentration of heavy metals in fish tissues? If the authors have these data showing the concentrations of metals in the tissues and organs of fish, they should include them in the results and then draw such a conclusion on this basis. Without that, the conclusion about deformations under the influence of metals is possible, but not confirmed until the presence or rather the absence of other causes of deformations in the tissues and organs of fish is established.

Author Response

Response to reviewer #3

Dear expert reviewer,

As authors of the manuscript now entitled “Impacts of open-pit gold mining on water quality and fish assemblages in the Santiago-Cayapas watershed, Esmeraldas, Ecuador”, we appreciated a lot your description and comments on the document, as we are certain and convinced, that they have been useful to eliminate obvious errors and to enrich the fluency and clarity of the entire article. Hereunder, we will reply and accomplish to change each mentioned point based on your input.

The authors have made an effort to evaluate in great detail the physico-chemical properties of water in mine-affected streams. Congratulations, well done.

However, this part of the title "Impact of open-pit gold mining on fish communities'', that part of the title is not confirmed by the text of the study.

The title about the impact of open-pit mining on fish is very brave, but based on this data it is not possible to say with certainty. This opens the door to speculation, which cannot be the basis for scientific arguments.

Has a microbiological analysis of the water samples been performed?

Microbiological analyses were not performed, there is a lack of researchers in this area in our institutions and alliances with international teams would be required for this purpose. However, the need of these analysis are briefly mentioned at the end of the discussion

Considering the fact that this is an area exposed to rainwater, bacteria and the pathogens that cause fish diseases can easily be washed away.

Has the health status of the fish been examined?

It was not done, in futures samples we will include analysis of stomach contents, which is something that could give us adequate information and is within the reach of our means

Has the autopsy of the fish been done?

 It was not done

Do the authors have data on the concentrations of metals and metallothioneins in fish?

We have some data, but we had problems with the contracted laboratory. At first, they sent the samples to Germany or Canada with good precision results. The following year they brought equipment to Ecuador and the first shipment of 48 fish presented erroneous results. Larger fish from the same site had fewer levels and even non-detection of metals detected months ago in smaller fishes of the same species. For the reality of our institution, it is also expensive, it costs 150 dollars to analyze 20 metals in a fish. However, it is part of studies that we would like to conduct once we have an alliance with an international team that is able to guarantee good results.

There is a very good statement on page 9: "There were significant differences in fish abundance among study sites."

Accordingly, the correct title would be:

Differences in fish abundance in rivers under the influence of open-pit gold mining in the Santiago-Cayapas basin, Esmeraldas, Ecuador.

Special remarks.

m3

Authors must cite literature according to the instructions for authors, consistently throughout the text. So in the case on page 2, "Jimenez-Prado"...or page 13, "Hammond et al. (2013)''

Use of punctuation, such as a period after 2006 (36) on page 2.

Spacing between words, such as on the page 2 ''(37, 38). The goal''..., or page 13 ''(57,44)''

Writing international units of measure m3 and not m3, mg O2L and not mg O2L; µScm-1, not µScm-1, etc.

Page 9, ‘’Under then’’ in ‘’Under them’’.

All done

The conclusions must be written in accordance with the facts from the paper. We cannot conclude that mining has reduced habitats, but that it has reduced the quality of those habitats. Based on the above, we cannot exclusively say that mines have changed fish communities? Because we have no data on the health status of the fish.

We agree to this observation. The term reduce habitat has been changed to restrict habitat, referring to water alterations that restrict certain conditions of these, which do not reduce their volume. The term assemblages refer to changes in structure or composition that do not necessarily involve community changes

Are body deformities in fish the only result of the effect of heavy metals on the fish organism? 

In the discussion and conclusions it is emphasized that there is a lack of concrete evidence to assert that the deformities are attributed to mining effluents. It may be due to the use of agrochemicals, an evidence approach is still necessary.

How can we tell if there are no data on the concentration of heavy metals in fish tissues? If the authors have these data showing the concentrations of metals in the tissues and organs of fish, they should include them in the results and then draw such a conclusion on this basis. Without that, the conclusion about deformations under the influence of metals is possible, but not confirmed until the presence or rather the absence of other causes of deformations in the tissues and organs of fish is established.

It is true, but the situation regarding the analysis of the concentration of metals was explained previously, the study should be improved considering this key variable

Once again and with all due respect, we are very thankful for your comments and corrections, which helped to see a few unclear parts and or even faults of our side within our manuscript. With your comments we were able to smooth the text, clarify missing parts or wrong spellings, which resulted to a much better than the initial version of this current study.

Thanks a lot on behalf of all authors

Round 2

Reviewer 3 Report

I thank the author for his detailed and honest answers to the questions. It is nice and scientific.

However, I regret that the authors are still not aware of the impropriety of their title.

The authors' change from "fish communities" to "fish assemblages" is a cosmetic change. Therefore, I have yet to explain in detail why I proposed a title that is consistent with what is in the text.

The authors' effort to examine the effects of mines on rivers is undeniable, and it is a very good job!

However, the authors' desire to show the effects of open-pit gold mining on fish communities/fish assemblages with what they have presented in their paper is not scientifically supported. Why not?

Because, as they honestly explain in their responses to my questions, the authors did not assess the health status of the fish sampled, nor did they conduct any tissue analysis. To make this clear, I will first use fact from the paper itself, sentences written by the authors: "The presence of individuals with body deformities is a clear indicator of deleterious effects of mine effluents on aquatic life, as there is ample evidence of histopathological damage and deformities in fish exposed to heavy metals.'' These authors concluded based on histopathological damage.

As our authors write in the responses regarding the analysis of concentrations of metals and metallothioneins in fish, ''We have some data, but we have had problems with the contracted laboratory.'' To put it simply, there are no such results.

So we do not know the concentrations of metals in fish, but we know their concentrations in rivers. But that is not enough to assess the effect of these metals on fish!

To explain to you and the authors what are the possible causes of the deformations, I send you this review paper

https://doi.org/10.1111/are.15125

Deformities in fish can be caused by a variety of reasons: chemical/physical water parameters, pH, low dissolved oxygen, herbicides, organophosphate and organochlorine pesticides, heavy metals, microbial diseases, etc. As we can see, metals are only one of the possible causes of deformations.

Given that the authors did not conduct a health survey, autopsy, or determine the metals in the tissues, on what basis do they conclude that the open-pit gold mining is affecting the health of the fish?

I repeat, it is an undeniable fact that the open-pit gold mining is affecting the quality of the water, because the authors have detected changes in the water, but without analysing the fish, they cannot draw any conclusion about the effects of the open-pit gold mining on the fish. For this reason, the changed title is also inappropriate, because the authors only noted the effects of the open-pit gold mining on water quality. And the second fact is that the authors in the paper did determine the number of fish communities in the rivers, which no one disputes. But it is questionable whether this number is the result of the impact of the open-pit gold mining, because this was not proved in the fish by anything, except the number and visual determination of body deformations (which, as you can see, can have very different causes).

This kind of title is unacceptable and I suggest again the title from my first review,

Differences in fish abundance in rivers under the influence of open-pit gold mining in the Santiago-Cayapas basin, Esmeraldas, Ecuador,

or closer to the title suggested by the author:

Differences in fish abundance in rivers under the influence of open-pit gold mining in the Santiago-Cayapas watershed, Esmeraldas, Ecuador.

Author Response

Dear expert reviewer, we are thankful for you further comments and input, which implies your deep conviction of improving our manuscript as much as possible. We will provide now a point-by-point response to your comments, as stated further below.

I thank the author for his detailed and honest answers to the questions. It is nice and scientific.

However, I regret that the authors are still not aware of the impropriety of their title.

The authors' change from "fish communities" to "fish assemblages" is a cosmetic change. Therefore, I have yet to explain in detail why I proposed a title that is consistent with what is in the text.

We gave it another thought and indeed it would be better to reform the title. So we agreed to change the title of our manuscript into “Differences in fish abundance in rivers under the influence of open-pit gold mining in the Santiago-Cayapas watershed , Esmeraldas, Ecuador”

In terms of the other comment of yours, we have used and applied the term assemblages according with the definition by Barletta, M., Dantas, D.V. (2016). Fish Assemblages. In: Kennish, M.J. (eds) Encyclopedia of Estuaries. Encyclopedia of Earth Sciences Series. Springer, Dordrecht. https://doi.org/10.1007/978-94-017-8801-4_138; where it is stated that a fish assemblage is simply a suite of species whose individuals are collected in the same area at the same time. Nonetheless, as we stated previously, we changed that title accordingly to your suggestion.

The authors' effort to examine the effects of mines on rivers is undeniable, and it is a very good job!

However, the authors' desire to show the effects of open-pit gold mining on fish communities/fish assemblages with what they have presented in their paper is not scientifically supported. Why not?

Because, as they honestly explain in their responses to my questions, the authors did not assess the health status of the fish sampled, nor did they conduct any tissue analysis. To make this clear, I will first use fact from the paper itself, sentences written by the authors: "The presence of individuals with body deformities is a clear indicator of deleterious effects of mine effluents on aquatic life, as there is ample evidence of histopathological damage and deformities in fish exposed to heavy metals.'' These authors concluded based on histopathological damage.

As our authors write in the responses regarding the analysis of concentrations of metals and metallothioneins in fish, ''We have some data, but we have had problems with the contracted laboratory.'' To put it simply, there are no such results.

It´s true, as we are editing another article, which includes the bioaccumulation of metals in fishes and macroinvertebrate analysis from previous data. We are right now working to obtain needed funds for a new monitoring and hereby to obtain and agreement with an accredited laboratory.

So we do not know the concentrations of metals in fish, but we know their concentrations in rivers. But that is not enough to assess the effect of these metals on fish!

To explain to you and the authors what are the possible causes of the deformations, I send you this review paper

https://doi.org/10.1111/are.15125

Deformities in fish can be caused by a variety of reasons: chemical/physical water parameters, pH, low dissolved oxygen, herbicides, organophosphate and organochlorine pesticides, heavy metals, microbial diseases, etc. As we can see, metals are only one of the possible causes of deformations.

 Thanks for the publication, we will consider that analysis we could do, since the traceability of pesticides is so expensive and we do not have experience or equipment for microbiology yet.

Given that the authors did not conduct a health survey, autopsy, or determine the metals in the tissues, on what basis do they conclude that the open-pit gold mining is affecting the health of the fish?

Well that is because we have been working almost a decade in the northern side of Esmeraldas Province (NW of Ecuador bordering with Colombia), where there are sectors that are exclusively mining sites and were there are no farms or plantations and the towns are just mining camps.

I repeat, it is an undeniable fact that the open-pit gold mining is affecting the quality of the water, because the authors have detected changes in the water, but without analysing the fish, they cannot draw any conclusion about the effects of the open-pit gold mining on the fish. For this reason, the changed title is also inappropriate, because the authors only noted the effects of the open-pit gold mining on water quality. And the second fact is that the authors in the paper did determine the number of fish communities in the rivers, which no one disputes. But it is questionable whether this number is the result of the impact of the open-pit gold mining, because this was not proved in the fish by anything, except the number and visual determination of body deformations (which, as you can see, can have very different causes).

This kind of title is unacceptable and I suggest again the title from my first review,

Differences in fish abundance in rivers under the influence of open-pit gold mining in the Santiago-Cayapas basin, Esmeraldas, Ecuador,

or closer to the title suggested by the author:

Differences in fish abundance in rivers under the influence of open-pit gold mining in the Santiago-Cayapas watershed, Esmeraldas, Ecuador.

It´s done as expected by you.

Once again, our gratitude with your effort and comments!